# Distribution of Pathogens in Early- and Late-Onset Sepsis Among Preterm Infants: A Decade-Long Study in a Tertiary Referral Neonatal Intensive Care Unit

**DOI:** 10.3390/jcm14010005

**Published:** 2024-12-24

**Authors:** Katarzyna Muszyńska-Radska, Krzysztof Szwed, Adrian Falkowski, Iwona Sadowska-Krawczenko

**Affiliations:** 1Department of Clinical Microbiology and Molecular Biology, Jan Biziel University Hospital No. 2, Ujejskiego 75, 85-168 Bydgoszcz, Poland; 2Center for Methodological Support of Scientific Research, Clinic of General and Minimally Invasive Surgery, Jan Biziel University Hospital No. 2, Ujejskiego 75, 85-168 Bydgoszcz, Poland; a.k.szwed@gmail.com (K.S.); izinizik@mat.umk.pl (A.F.); 3Department of Neonatology, Faculty of Medicine, Ludwik Rydgier Collegium Medicum in Bydgoszcz, Nicolaus Copernicus University in Torun, Ujejskiego 75, 85-168 Bydgoszcz, Poland; iwonasadowska@cm.umk.pl

**Keywords:** EOS, LOS, sepsis, bloodstream infection, newborn, preterm infants, pathogens

## Abstract

**Background:** Neonatal sepsis, a severe infection in newborns, remains one of the leading causes of morbidity and mortality among preterm infants. This study aimed to investigate the distribution of pathogens responsible for early-onset sepsis (EOS) and late-onset sepsis (LOS), the annual variability of pathogens responsible for each type of infection, and potential trends in their profiles in preterm infants from a tertiary care neonatal intensive care unit over a ten-year period. **Methods:** We analyzed 177 episodes of confirmed bloodstream infection between 1 January 2014 and 31 December 2023. An episode of confirmed bloodstream infection was defined by the growth of a single potential pathogen in the blood of an infant who met four criteria: showing clinical symptoms of infection, having abnormal hematologic parameters, receiving appropriate antibiotics for less than 5 days, and being born before 37 weeks of gestation. Pathogens were isolated from aseptically collected blood samples, processed within 2 h, incubated using the BACTEC system, and identified by Gram stains and selective media cultures. Pathogen identification was performed using standardized biochemical tests or MALDI-TOF MS. Sepsis was classified as EOS if it occurred within the first 72 h of life and as LOS if it occurred after 72 h. **Results:** Among the confirmed bloodstream infections, EOS accounted for 31 cases, while LOS accounted for 146 cases. Escherichia coli was identified as the primary pathogen responsible for early-onset sepsis (EOS), while Coagulase-negative Staphylococcus (CoNS) was most commonly associated with late-onset sepsis (LOS). The differences in the prevalence of these bacteria between EOS and LOS were statistically significant. However, no significant differences were found in the distribution of pathogens across different years, nor were there significant trends in their frequency over the study period. Our results demonstrate significant stability in the distribution of pathogens causing sepsis over the ten-year observation period, even during the COVID-19 pandemic. **Conclusions:** Understanding the temporal distribution of pathogens in neonatal sepsis can help prevent the overuse of antibiotics and support the implementation of screening programs, empiric therapy, and strategies to prevent healthcare-associated infections.

## 1. Introduction

Sepsis is one of the leading causes of death globally. According to the World Health Organization (WHO), data published in 2020 indicated there were 48.9 million cases and 11 million sepsis-related deaths worldwide, accounting for 20% of all global deaths. Nearly 20 million of these cases occurred in children under the age of five [1]. Neonatal sepsis is one of major causes of death and prolonged length of hospital stay during neonatal period, especially in preterm infants. Diagnosis neonatal sepsis is challenging for neonatologists due to its variable, nonspecific signs and symptoms [2]. Early diagnosis, rapid intervention, and comprehensive preventive strategies are crucial to minimize adverse outcomes. Contemporary prevention strategies focus on a multifaceted approach, including improving maternal care, enhancing infection control through strict hygiene protocols, judicious use of antibiotics, development of rapid diagnostic techniques, immunomodulatory interventions, and the implementation of screening tools and early warning systems [3]. Sepsis is a bloodstream infection caused by bacteria, viruses or fungi, and it is associated with hemodynamic changes and severe systematic clinical manifestations. The pathogenesis of neonatal sepsis is complex, involving interactions between the immature immune system, pathogens, and the body’s defense mechanisms. The identification of pathogens responsible for sepsis and the determination of their antibiotic sensitivity are critical for the effective management of bloodstream infections. This process directly influences the success of the therapy, as well as the duration of hospitalization and associated costs [2,4].

Neonatal sepsis is categorized based on the timing of symptom onset into two types: early-onset sepsis (EOS) and late-onset sepsis (LOS). According to WHO, EOS occurs within the first three days of life (<72 h), although some research extends this period to the first seven days of life [5]. LOS occurs at or after 72 h of life. EOS is typically caused by organisms transmitted vertically before or during birth, through contaminated amniotic fluid or during vaginal delivery by organisms that colonize or infect the mother’s genitourinary system [6]. Late-onset sepsis (LOS) mostly occurs due to the transmission of pathogens from the hospital environment and over environmental sources. It is associated with invasive therapeutic procedures that can facilitate this transmission. LOS can also be a late manifestation of a vertically transmitted infection [6,7].

Determining the epidemiology of sepsis amid rising antibiotic resistance is vital for improving neonatal care, reducing mortality and long-term complication [8].

The primary objective of this study was to investigate the annual variability and ten-year trend in the distribution of microorganisms responsible for EOS and LOS in the neonatal intensive care unit. The secondary objective was to analyze the statistical associations between the pathogens involved in EOS and LOS.

## 2. Materials and Methods

### 2.1. Study Design

This study is a retrospective analysis of positive blood cultures from hospitalized neonates with confirmed sepsis who met the study’s inclusion criteria. The database covers the period from 1 January 2014 to 31 December 2023 and includes preterm infants (<37 weeks) hospitalized in the Neonatal Intensive Care Unit at Jan Biziel University Hospital in Bydgoszcz.

### 2.2. Definition

Early-onset sepsis (EOS) is defined as an infection that develops within the first 72 h of life, whereas late-onset sepsis (LOS) develops after 72 h of life.

### 2.3. Inclusion Criteria

An episode of confirmed bloodstream infection with positive blood culture was defined as the growth of single potential pathogen from the blood of an infant who met four of the following criteria:The infant presented clinical symptoms of infection: respiratory distress, apnea, tachycardia or bradycardia, systemic hypotension or hypoperfusion, hypothermia or fever, convulsion, hypotonia, irritability or lethargy, feeding intolerance or intestinal obstruction, neonatal jaundice, hypoglycemia.One or more abnormal hematologic parameters; white blood cell count, increased immature/ total neutrophil count, elevated C-reactive protein level, abnormal procalcitonin level.Appropriate antibiotics were used for at least 5 days (especially in the case of Coagulase-negative Staphylococci [CoNS] bacteremia) or for less than 5 days if death occurred while receiving antibiotics.The infant’s gestational was less than 37 weeks.

Repeatedly isolated pathogens were considered as part of the same episode if the period between samples was less than 7 days for all organisms, except for CoNS and fungal infection, where the period was 10 days.

We excluded samples at risk of contamination, which can occur due to improper skin disinfection (crucial when suspecting CoNS infections), tainting of the blood culture bottles, or improper handling of collected material. The criteria for contamination were as follows:Isolation of bacteria usually considered contaminants (e.g., *Bacillus* spp., *Micrococcus* spp. *Corynebacterium* spp.);Isolation of mixed flora of CoNS;Isolation of CoNS in the absence of infection symptoms, as considered by the neonatologist.

Patients who did not fulfill the four inclusion criteria and those who were under suspicion for contamination were not included in the analysis.

### 2.4. Methodology of Sample Collection and Microbiological Techniques

Blood samples were collected by a neonatologist. The skin was disinfected with 0.5% chlorhexidine for 1 min. Approximately one milliliter of blood was aseptically collected from a vein into PEDS Plus /F culture vials from Becton Dickinson. One blood culture bottle was collected and delivered to the microbiology laboratory within 2 h. Bottles were incubated in the BACTEC automated blood culture system for seven days at 35 ± 2 before reporting no growth. Positive vials were identified by performing Gram stains and culturing in selective media. Identification was performed using a standardized biochemical tests and the MALDI TOF MS (Matrix-Assisted Laser Desorption/Ionization-Time Of Flight- Mass Spectrometry) method based on mass spectrometry for microorganism identification. Antimicrobial susceptibility testing was performed on all isolates using the standard disk-diffusion method on agar plates. The results were interpreted according to the recommendations of the European Committee on Antimicrobial Susceptibility Testing (EUCAST).

### 2.5. Treatment Protocol

EOS treatment protocol included ampicillin and aminoglycoside. LOS treatment included vancomycin plus piperacillin-tazobactam or meropenem. Upon receiving the microbiological results, the therapy was adjusted based on the antimicrobial susceptibility of the isolated strain causing the infection.

### 2.6. Statistical Analysis

The statistical analyses included chi-square and Fisher’s exact tests, applied as appropriate, to compare the prevalence of specific microorganisms between EOS and LOS groups. Additionally, temporal trends in microorganism prevalence across years were examined using Kruskal–Wallis and Kendall’s tau tests. The Kruskal–Wallis test assessed differences in microorganism presence across distinct years, while Kendall’s tau was employed to evaluate correlations between year and infection rate trends. To adjust for multiple comparisons, Bonferroni correction was applied to the *p*-values, with significant values set to *p* < 0.05.

## 3. Results

During the study period from 1 January 2014 to 31 December 2023, we registered 317 episodes of positive blood cultures. However, 141 cases were excluded from final analysis because they did not fulfill the inclusion criteria for confirmed bloodstream infection or were suspected of contamination. A total of 177 episodes of positive blood culture with confirmed bloodstream infection, represented by 163 preterm neonates, were included in the analysis. Among them, 147 patients had one episode of confirmed sepsis with positive blood culture during hospital stay, while 16 had more than one episode. Early-onset sepsis was found in 31 cases, and late-onset sepsis was found in 146 cases.

### 3.1. Pathogens Involved in Early-Onset Sepsis

Among preterm infants with EOS, the leading etiological agent identified in the blood cultures was a Gram-negative organism accounting for 25 cases (80.6%). The majority of isolates were *Enterobacterales* family rods with *Escherichia coli* dominating in 18 cases (58%). Gram-positive organisms accounted for six cases (19.3%) represented by *Streptococcus agalactiae*, *Listeria monocytogenes*, and CoNS. Table 1 shows distribution of pathogens responsible for EOS.

### 3.2. Pathogens Involved in Late-Onset Sepsis

The most common pathogens in LOS were Gram-positive organisms. Among them, Coagulaso-negative *Staphylococcus* (CoNS) was the leading pathogen, accounting for 81 cases (55.4%), followed by *Staphylococcus aureus*, which accounted for 12 cases (8.2%), *Enterococcus* spp. accounting for 6 cases (4.1%), and *GBS* accounting for 3 cases (2%). Gram-negative organisms were identified in 30 cases (20.6%), particularly represented by *Enterobacterales* family rods in 28 cases (19,1%). The most frequent Gram-negative organisms were *Klebsiella* spp. in 12 cases (8.2%) and *Escherichia coli* in 8 cases (5.4%). Fungi represented 14 of LOS cases (9.6%), with *Candida albicans* accounting for 8 cases (5.5%) and other *Candida* spp. for 6 cases (4.1%). Table 1 shows the distribution of pathogens responsible for LOS.

Throughout the ten-year study period, no significant year-to-year differences were observed in the percentage distribution of pathogens in either EOS or LOS. Table 2 and Table 3 show the annual distribution of pathogens associated with both types of sepsis.

## 4. Discussion

This study showcases the distribution of pathogens associated with EOS and LOS over a decade, including the COVID-19 pandemic year (2020–2023). Our analyses revealed significant differences in the pathogen etiology of EOS and LOS and identified the microorganisms most commonly responsible for both types of infections.

No statistically significant differences were found in the annual variability of pathogens causing each type of infection. Additionally, no trends were observed in the annual profile over the ten-year observation period.

The stable distribution of pathogens during the COVID-19 pandemic, observed in our study, presents an intriguing epidemiological topic. Most studies on the pandemic’s impact have focused on adult intensive care units, not neonatal ones. This might be partly because neonates were less affected by SARS-CoV-2 and did not experience the same complications, resulting in limited data [9].

It might be expected that stricter hygiene and sanitary measures would reduce infection and change the microbial profile. However, the pandemic showed that enhanced infection control measures do not always reduce infections [10,11].

In neonatal intensive care units, the increased risk of hospital-acquired infections during the pandemic could be due to reduced skin-to-skin contact with mothers, disruptions in family-centered care, which is beneficial for the development neonates, and frequent understaffing [12].

Psychosocial factors among medical personnel also play a significant role. Continuous stress, prolonged epidemic threat, and the use of personnel protective equipment can paradoxically reduce the effectiveness of sanitary procedures [11]. Fatigue, distraction, and the need for quick decisions under chronic stress can lead to lapses in sanitary regimes [10].

In our study, we investigated distribution of pathogens in EOS and LOS in preterm infants. The main pathogen causing EOS in this group was *Escherichia coli*, similar to findings by Stoll B.J. et al. [13]. Stoll B.J. et al. emphasized the rising infection rates of *Escherichia coli* among preterm infants. Newborns with EOS infected with Gram-negative bacteria such as *Escherichia coli* are more likely to die. Research by Lee S.M. et al. confirms the mortality associated with this type of sepsis [5,13]. *Streptococcus agalactiae* (GBS) remains a significant etiological factor in early-onset sepsis (EOS). In our study, GBS was less prevalent than *Escherichia coli* in preterm neonates, similar to the findings reported by Stoll et al. and Liu et al. [13,14]. The reduced number of cases may indicate the effectiveness of screening pregnant women and the implementation of intrapartum antibiotic prophylaxis (IAP), which should continue to be adhered to and monitored [13]. The standard empiric therapy with ampicillin and aminoglycoside may be insufficient in the context of the increasing prevalence of *E. coli* as the main pathogen causing early-onset sepsis (EOS) in preterm neonates. While Group B Streptococcus (GBS) is generally susceptible to ampicillin, E. coli can be resistant to both ampicillin and aminoglycosides. Therefore, further analyses of the antibiotic susceptibility of the strains and continuous monitoring of the rising resistance to these drugs are essential. It is necessary to establish recommendations that include antibiotics with a broader spectrum, effective in the appropriate clinical situations [13]. During our 10-years study period, we found two cases of early-onset sepsis caused by CoNs that met our inclusion criteria. The role of CoNS in early-onset sepsis is debatable. CoNS may not always indicate true infection, as some positive cultures might result from contamination or catheter colonization. Various studies report differing rates of EOS due to CoNS. These differences likely arise from variations in the study methodologies. For example, Mariani M. et al. considered any positive culture as a case of EOS while Stoll et al. classified CoNS as contamination unless it was isolated in more than two separate cultures and the infant received more than five days of appropriate antibiotics [13,15]. Obtaining an adequate volume of blood and a sufficient number of cultures is crucial for interpreting blood cultures result. A minimum volume of 1 mL of blood and two or more blood cultures bottles are most advisable. However, obtaining these samples from preterm infants can be challenging, often resulting in only a single positive blood culture, as seen in our study [16].

The most common pathogen in LOS was *Coagulaso-negative Staphylococcus* (CoNS). In most studies, CoNS dominate as pathogens in LOS, but percentages differ between studies. Similar CoNS predominance was reported by Kostlin-Gille N. et al. [17]. In some studies, like in Dustin D. et al., sepsis caused by CoNS was similar to sepsis caused by *S. aureus* [18]. In contrast, *Klebsiella pneumonia* was the most common pathogen in a study by Liu J. et al. [14]. Another significant Gram-positive etiological agent in preterm infants was *S. aureus*. Gram-negative organisms were mainly represented by *Enterobacterales* family rods, especially *Klebsiella* spp. and *Escherichia coli.* Similar results were found in studies by Dustin D. et al. and Kostlin-Gille N. et al. [17,18]. Fungal infections are also important causes of late-onset sepsis in preterm infants. The incidence rate of fungal infection in newborns was reported to be 9.6%. Other studies, such as those by Vergano S. et al. and Lee M.S. et al., confirm this percentage. The most common fungus is *Candida albicans*. According to Lee M.S. et al., fungi are significant etiological agents associated with high mortality in LOS [5,19].

Empiric therapy using vancomycin and piperacillin-tazobactam can be effective against late-onset infections caused by CoNS, *S. aureus*, and Gram-negative rods. However, continuous monitoring and analysis of the antimicrobial susceptibility of these microorganisms are essential. Carbapenems should be reserved for specific situations, particularly in the case of epidemic occurrences of Gram-negative bacteria producing extended-spectrum beta-lactamases (ESBL). These findings are also supported by the studies of Mariani M. et al. The selection of antibiotics for late-onset infections should be based on retrospective analyses of the etiology and antibiotic resistance of the microorganisms causing these infections in the specific ward. This underscores the importance of conducting studies like ours [15,20].

Neonatal sepsis remains a significant cause of morbidity and mortality. Identifying specific pathogens allows for selecting the most effective antibiotic therapy, which is crucial for updating guidelines on sepsis prophylaxis and empirical treatment. Understanding the potential pathogens involved in neonatal sepsis can help prevent the overuse of antibiotics, thereby reducing the risk of resistance [13,18,21]. Identification of early-onset sepsis pathogens may indicate the need for improvement in perinatal care [7,19]. Knowledge of late-onset sepsis pathogens helps refine procedures in neonatal intensive care units, minimizing the risk of nosocomial infections. Our efforts should focus on activities such as continuous hand washing training. Additionally, there is a constant need for training personnel on proper blood culture collection techniques, as this is one of the main causes of contamination [8]. Pathogen research stimulates the development of faster and more sensitive diagnostic tests, crucial for early sepsis detection. Identifying the most common pathogens can guide research into new vaccines for pregnant women or high-risk newborns [13].

## 5. Conclusions

Despite the ten-year observation period, which included the COVID-19 pandemic, we did not find significant annual differences in the pathogen profiles associated with EOS and LOS. We observed that the differences in bacterial prevalence between early-onset sepsis (EOS) and late-onset sepsis (LOS) were statistically significant. The most common organism causing EOS was *Escherichia coli*, while LOS was predominantly caused by Coagulase-negative staphylococci (CoNS). Understanding the epidemiology of sepsis, especially in the context of increasing antibiotic resistance, is crucial for enhancing neonatal care and reducing mortality rates.

## Figures and Tables

**Table 1 jcm-14-00005-t001:** Distribution of pathogens in EOS and LOS.

Pathogen	EOS [n = 31]		LOS [n = 146]		Corrected *p*-Value
	n	%	n	%	
**Gram-positive**	6	19.35	102	69.86	0.004
CoNS	2	6.45	81	55.48	0.018
* Staphylococcus aureus*	0	0	12	8.22	1.000
* Enterococcus* spp.	0	0	6	4.11	1.000
* Streptococcus agalactaie*	2	6.45	3	2.05	1.000
* Listeria monocytogenes*	2	6.45	0	0	0.540
**Gram-negative**	25	80.65	30	20.55	0.003
** * Enterobacterales* **	23	74.19	28	19.18	0.004
* Escherichia coli*	18	58.06	8	5.48	0.018
* Klebsiella* spp.	2	6.45	12	8.22	1.000
* Enterobacter cloacae*	0	0	3	2.05	1.000
* Proteus mirabilis*	0	0	2	1.37	1.000
* Morganella morganii*	1	3.23	0	0	1.000
* Serratia marcescens*	0	0	1	0.68	1.000
* Citrobacter* spp.	1	3.23	2	1.37	1.000
* Pantea agglomerans*	1	3.23	0	0	1.000
** Other Gram-negative**	2	6.45	2	1.37	0.568
* Acinetobacter* spp.	0	0	1	0.68	1.000
* Pseudomonas aeruginosa*	1	3.23	1	0.68	1.000
* Haemophilus influenzae*	1	3.23	0	0	1.000
**Fungi**	0	0	14	9.59	0.536
* Candida albicans*	0	0	8	5.48	1.000
other *Candida*	0	0	6	4.11	1.000

The table presents the microorganisms responsible for both types of sepsis. Our analyses revealed notable differences in the pathogen profiles between early-onset sepsis (EOS) and late-onset sepsis (LOS).

**Table 2 jcm-14-00005-t002:** Annual distribution of pathogens associated with early-onset sepsis.

EOS										
	2014	2015	2016	2017	2018	2019	2020	2021	2022	2023
Number of hospitalizations	438	433	772	810	752	759	645	707	680	635
**Gram-positive**	0	0	1	0	1	0	2	2	0	0
CoNS	0	0	0	0	0	0	2	0	0	0
* Staphylococcus aureus*	0	0	0	0	0	0	0	0	0	0
* Enterococcus* spp.	0	1	0	0	0	0	0	0	0	0
* Streptococcus agalactaie*	0	0	1	0	1	0	0	0	0	0
* Listeria monocytogenes*	0	0	0	0	0	0	0	2	0	0
**Gram-negative**	2	5	5	3	2	3	0	1	2	2
** * Enterobacterales* **	2	4	4	3	2	3	0	1	2	2
* Escherichia coli*	2	4	3	3	1	1	0	1	2	1
* Klebsiella* spp.	0	0	0	0	0	2	0	0	0	0
* Enterobacter cloacae*	0	0	0	0	0	0	0	0	0	0
* Proteus mirabilis*	0	0	0	0	0	0	0	0	0	0
* Morganella morganii*	0	0	0	0	1	0	0	0	0	0
* Serratia marcescens*	0	0	0	0	0	0	0	0	0	0
* Citrobacter* spp.	0	0	1	0	0	0	0	0	0	0
* Pantea agglomerans*	0	0	0	0	0	0	0	0	0	1
** Other Gram-negative**	0	1	1	0	0	0	0	0	0	0
* Acinetobacter* spp.	0	0	0	0	0	0	0	0	0	0
* Pseudomonas aeruginosa*	0	0	1	0	0	0	0	0	0	0
* Haemophilus influenzae*	0	1	0	0	0	0	0	0	0	0
**Fungi**	0	0	0	0	0	0	0	0	0	0
* Candida albicans*	0	0	0	0	0	0	0	0	0	0
Other *Candida*	0	0	0	0	0	0	0	0	0	0

This table presents the annual number of isolations of each pathogen associated with early-onset sepsis (EOS). There were no years that significantly differed from the others, nor were there any significant trends in the percentage of infections over the years.

**Table 3 jcm-14-00005-t003:** Annual distribution of pathogens associated with late-onset sepsis.

LOS										
	2014	2015	2016	2017	2018	2019	2020	2021	2022	2023
Number of hospitalization	438	433	772	810	752	759	645	707	680	635
**Gram-positive**	8	11	18	9	10	12	6	14	7	6
CoNS	7	10	13	6	10	12	3	10	4	6
* Staphylococcus aureus*	0	1	2	1	0	0	3	2	2	0
* Enterococcus* spp.	1	0	1	2	0	0	0	1	1	0
* Streptococcus agalactaie*	0	0	2	0	0	0	0	1	0	0
* Listeria monocytogenes*	0	0	0	0	0	0	0	0	0	0
**Gram-negative**	4	0	3	3	3	2	2	6	6	1
** * Enterobacterales* **	3	0	3	3	3	2	2	6	6	0
* Escherichia coli*	1	0	1	1	1	2	1	1	0	0
* Klebsiella* spp.	0	0	1	0	2	0	0	4	5	0
* Enterobacter cloacae*	1	0	1	1	0	0	0	0	0	0
* Proteus mirabilis*	0	0	0	1	0	0	1	0	0	0
* Morganella morganii*	0	0	0	0	0	0	0	0	0	0
* Serratia marcescens*	1	0	0	0	0	0	0	0	0	0
* Citrobacter* spp.	0	0	0	0	0	0	0	1	1	0
* Pantea agglomerans*	0	0	0	0	0	0	0	0	0	0
** Other Gram-negative**	1	0	0	0	0	0	0	0	0	1
* Acinetobacter* spp.	1	0	0	0	0	0	0	0	0	0
* Pseudomonas aeruginosa*	0	0	0	0	0	0	0	0	0	1
* Haemophilus influenzae*	0	0	0	0	0	0	0	0	0	0
**Fungi**	2	1	1	1	2	0	1	1	2	3
* Candida albicans*	1	0	1	1	2	0	0	1	0	2
Other *Candida*	1	1	0	0	0	0	1	0	2	1

This table presents the annual number of isolations of each pathogen associated with late-onset sepsis (LOS). There were no years that significantly differed from the others, nor were there any significant trends in the percentage of infections over the years.

## Data Availability

The data presented in this study are available on request from the corresponding author.

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
