# Peer review of "Distribution of Pathogens in Early- and Late-Onset Sepsis Among Preterm Infants: A Decade-Long Study in a Tertiary Referral Neonatal Intensive Care Unit"

_jcm, 2024, doi:10.3390/jcm14010005_

Round 1
Reviewer 1 Report
Comments and Suggestions for Authors
This manuscript presents a valuable 10-year retrospective analysis of pathogen distribution in early-onset sepsis (EOS) and late-onset sepsis (LOS) among preterm infants. The study provides important epidemiological data that can inform clinical practice and antibiotic stewardship programs.
Major Strengths:
- Long study period (10 years) with consistent methodology
- Clear inclusion criteria and contamination definitions
- Rigorous microbiological techniques
- Detailed statistical analysis with appropriate corrections for multiple comparisons
- Inclusion of data during the COVID-19 pandemic period
Specific comments:
Introduction:
- At the beginning of this section, authors should provide more background on the global burden of sepsis (doi: 10.3390/epidemiologia5030032).
- Include more recent references on current prevention strategies
Methods:
- Clarify the rationale for using 72 hours as the EOS/LOS cutoff
- Provide more details about the antibiotic therapy protocols used
Results:
- Add information about patient demographics
- Include data on length of hospital stay
Discussion:
- Expand on the implications of stable pathogen distribution during COVID-19
- Address the relatively low number of GBS cases
- Discuss the implications for empiric antibiotic choices
Reviewer 2 Report
Comments and Suggestions for Authors
I have read with great interest the manuscript by Muszynska-Radska, which details a 10-year study in a neonatal ICU examining the distribution of pathogenic organisms in early- and late-onset neonatal sepsis.
The introduction is well-written, and I do not have significant recommendations for changes. However, upon reviewing the manuscript and results, it appears that the authors have employed statistical tests including p-values to analyze differences between pathogens associated with early- versus late-onset sepsis. This approach does not align with the primary objective stated in lines 58–60. To address this, I suggest adding a secondary objective at the end of the introduction, specifying that the study also aims to establish statistical associations between pathogens involved in early- vs late-onset sepsis.
Regarding the methodology, Section 2.3 on Inclusion Criteria requires further clarification. Specifically, in lines 97–98, how were patients suspected of contamination identified? What criteria were used to determine whether a positive blood culture was a contaminated sample? Was this based on specific clinical criteria, or was it left to the discretion of the clinician? Providing more detail here would enhance the transparency of the study.
In line 165 of the discussion, the authors should clarify what is meant by the "pandemic period" by explicitly referring to it as the COVID-19 pandemic period. Additionally, I agree with the authors' statement in lines 185–186 regarding the advisability of obtaining a minimum of 1 mL of blood and using two or more blood culture bottles. However, this assertion would benefit from a supporting reference to strengthen its credibility.
In the discussion, the authors note that no significant trend was observed in the annual profile of pathogenic organisms over the 10-year study period. I would recommend expanding on this point by discussing whether any trends were expected, and if so, what trends might have been anticipated. Furthermore, the authors should consider exploring whether they expected any changes in pathogen distribution during the COVID-19 pandemic year. Addressing these points would add greater depth and insight to the discussion.
